# A Comprehensive Risk Factor Exploration: Korean Nationwide Cross-Sectional Study of Sarcopenia Obesity in Young-Old Males—Investigating the Prevalence, Somatometric, Biochemical, and Behavioral Traits

**DOI:** 10.3390/healthcare12060700

**Published:** 2024-03-21

**Authors:** Jongseok Hwang, Soonjee Park

**Affiliations:** 1Institute of Human Ecology, Yeungnam University, Gyeongsan 38541, Republic of Korea; sfcsfc44@naver.com; 2Department of Clothing and Fashion, Yeungnam University, Gyeongsan 38541, Republic of Korea

**Keywords:** sarcopenic obesity, clinical risk factors, prevalence, odds ratio

## Abstract

Sarcopenic obesity refers to the coexistence of sarcopenia and obesity. This study assessed the prevalence of and risk factors for sarcopenic obesity in community-dwelling older males in Korea. This cross-sectional study analyzed data from 1060 community-dwelling participants. The participants were screened for sarcopenia and obesity. This study examined various risk factors, composed of somatometric, biochemical, and behavioral traits, including age; height; weight; body mass index; waist circumference; skeletal muscle mass index; fasting glucose, triglycerides, and cholesterol levels; systolic and diastolic blood pressure; and smoking and drinking traits. The prevalence of sarcopenic obesity in men had a weighted value of 9.09% (95% CI: 7.20–11.41). The clinical risk factors included short height, as well as high weight, body mass index, waist circumference, skeletal muscle mass index, triglyceride levels, systolic blood pressure, and diastolic blood pressure. Specific prevalence and clinical risk factors for sarcopenic obesity were found among community-dwelling older men. These findings may be beneficial for primary care clinicians and healthcare professionals for identifying individuals with sarcopenic obesity and referring them for early detection and treatment.

## 1. Introduction

Sarcopenic obesity is a condition in which an individual exhibits both sarcopenia and obesity. Sarcopenia is characterized by the progressive loss of muscle mass, strength, and function in older adults [1]. Obesity is the accumulation of excess body fat that can have a negative impact on overall health, including an increased risk of chronic diseases, such as diabetes, hyperlipidemia, hypertension, and cardiovascular disease [2]. The coexistence of obesity and low muscle mass can exacerbate health problems, contribute to the development of chronic degenerative diseases and disabilities, prolong hospital stays, and impose a significant burden on the sustainability and effectiveness of healthcare services, social welfare, public health, and policymaking [3]. Several studies have demonstrated that individuals with sarcopenic obesity experience greater morbidity, disability, and mortality than those with low muscle mass or obesity alone [3,4].

The aging population in Asia is growing rapidly. Similarly, Korea’s demographic landscape is undergoing a swift transformation, marked by a notable acceleration in the proportion of the older population. It is anticipated that by 2050, the population of older people (over 65 years) in Korea will increase significantly, accounting for 40% (19 million) of the total population, representing a substantial increase from the current 15% (5 million) in 2022 [5]. Moreover, the prevalence of obesity among Koreans aged at least 65 years has increased. Nam et al. [6] found that the prevalence of obesity among men increased from 28.0% in 2009 to 34.8% in 2018. This demographic shift raises concerns about age-related complications, including sarcopenia and obesity, which could have significant health consequences for Korean society.

Despite the potential negative consequences and the growing population of older people, healthcare professionals and primary care clinicians face limitations in their ability to diagnose sarcopenic obesity owing to insufficient knowledge and diagnostic tools. The average time allotted per patient visit by general practitioners is less than 10 min, which necessitates that primary care clinicians identify the probability of a patient having sarcopenic obesity before considering a referral for diagnosis and treatment. Moreover, a deficiency in clinicians’ awareness of sarcopenic obesity as a disease increases the chance of missed diagnosis [7]. Understanding the characteristics of the key risk factors related to early detection and prevention is crucial to effectively address this challenge [8]. The early detection of symptomatic individuals can significantly impact timely diagnosis and intervention. Delayed or missed diagnosis of sarcopenic obesity can lead to complications such as poor functional recovery, decreased quality of life, and waste of government healthcare resources.

Most sarcopenic obesity studies group participants into a single category, disregarding the various changes in protein homeostasis, epigenetic modifications, immune responses, and metabolic regulation by age group [9]. Therefore, to identify and assess the characteristics of sarcopenic obesity according to age, it is essential to classify older individuals according to age group; young-old (aged 65–74), old (aged 75–84), and oldest-old (aged 85 years and above) [5,10,11,12,13].

This study assessed the prevalence of sarcopenic obesity and identify its associated factors in young-old male adults aged 65–74 years. Two hypotheses were presented: a specific prevalence of sarcopenic obesity among community-dwelling older men exists, and specific risk factors and their odds ratios are associated with sarcopenic obesity.

## 2. Materials and Methods

### 2.1. Participants and Study Population

The Korea National Health and Nutrition Examination Survey (KNHANES) is conducted annually by the Ministry of Health and Welfare and the Korea Institute for Health and Social Affairs. The present research data were collected from 37,573 KNHANES participants between 2008 and 2011. After excluding 35,736 participants, who were either female or males aged less than 65 or more than 74 years old, 1837 participants were included in this study. Out of these, 777 participants were excluded due to either not undergoing dual-energy X-ray absorptiometry (DEXA) examination or failing to respond to the health survey. This resulted in a final sample of 1060 male participants aged 65–74 years. Among this final sample, 95 individuals were classified as having both sarcopenia and obesity, while 965 were classified as normal. This cross-sectional study was approved by the Institutional Review Board of the Center for Disease Control and Prevention (approval numbers 2008-04EXP-01-C, 2009-01CON-03-2C, 2010-02CON-21-C, and 2011-02CON-06-C). All individuals in this study provided informed consent prior to participation in accordance with ethical guidelines.

### 2.2. Diagnosis of Sarcopenic Obesity

The two components of sarcopenic obesity, sarcopenia and obesity, are defined as follows. Sarcopenia is characterized by low skeletal muscle mass, calculated as the sum of the appendicular skeletal muscle mass (ASM). Skeletal muscle mass in the limbs was assessed using dual-energy X-ray absorptiometry (DEXA) with QDR4500A equipment manufactured by Ho-logic, Inc., Bedford, MA, USA. To evaluate muscle mass, this study employed the ASM (kg)/BMI (kg/m^2^) ratio, referred to as the Skeletal Muscle Mass Index (SMI). The diagnostic criterion for sarcopenia was an SMI value below 0.789 in males, according to the guidelines established by the Foundation for the National Institutes of Health Sarcopenia Project [14].

Obesity is characterized by excessive fat accumulation that adversely affects individual health, and is identified by a BMI ≥ 25 kg/m^2^ and central obesity, defined as a waist circumference (WC) greater than 90 cm in men and 80 cm in women in the Asian population by the World Health Organization [15].

### 2.3. Variables

#### 2.3.1. Somatometric Variables

Somatometric variable data were collected from all participants. To ensure measurement precision, the participants were instructed to remove footwear, socks, headwear, and hairpins, and wear lightweight clothing. Height and weight were measured meticulously using calibrated automatic body measurement equipment, with accuracy recorded to the nearest 0.1 cm or kg. Following these measurements, the body mass index (BMI) was computed by dividing weight (kg) by the square of height (m^2^). Additionally, waist circumference (WC) was measured with precision to the nearest 0.1 cm in a horizontal plane at the midline between the last rib and the iliac crest, following a normal expiration [16].

#### 2.3.2. Biochemical and Blood Pressure Variables

Biochemical variables, including fasting glucose (FG), triglycerides, and total cholesterol (TC) levels, were analyzed using the LABOSPECT 008AS platform (Hitachi High-Tech Co., Tokyo, Japan). Blood samples were drawn from the non-dominant arm after an overnight fast of at least eight hours. After collection, the blood was mixed with a coagulation promoter and centrifuged using a mobile examination vehicle. All tests were performed within 24 h of blood sample collection. Systolic blood pressure (SBP) and diastolic blood pressure (DBP) were measured by a trained practitioner using a mercury sphygmomanometer. The blood pressure cuff was positioned at the heart level with the participants in a seated position after a minimum of five minutes of rest [16,17].

#### 2.3.3. Behavioral Traits

Information regarding the participants’ behavioral habits, including smoking and drinking, was obtained through a survey. The participants were categorized into three groups based on their cigarette smoking and alcohol drinking habits: non-users, ex-users, and current users. These measured variables play a pivotal role in assessing diverse aspects of health and their potential contribution to diseases within a study population [16].

### 2.4. Statistical Analysis

Data analysis was conducted using the SPSS software (version 25.0; SPSS Inc., Chicago, IL, USA). To reflect the entire older population in Korea, weighted values were applied to each sample through a three-step process: (1) calculation of the base weight, (2) adjustment for non-responses, and (3) post-stratification adjustment to match the entire previous census population. All analyses were conducted using complex sampling, with each individual adjusted for the weights provided by the KNHANES. Parametric variables were compared between the sarcopenic obesity and control groups using independent *t*-tests, whereas nonparametric variables were analyzed using the chi-square test. Covariate-adjusted multiple logistic regression analysis was used to predict sarcopenic obesity and calculate the odds ratios associated with sarcopenic obesity risk factors. In order to mitigate the potential for inflated Type I error rates due to multiple comparisons, we implemented the Bonferroni correction method.

## 3. Results

### 3.1. Prevalence

The prevalence of sarcopenic obesity is presented in Table 1. The weighted prevalence was 9.09% (95% confidence interval (CI): 7.20–11.41). The unweighted prevalence was 8.95%, and the prevalence of normal weight was 90.91% (95% CI: 88.59–92.80) (Table 1).

### 3.2. Risk Factors

#### 3.2.1. Somatometric Variables

Statistically significant differences in somatometric variables were observed for height, weight, BMI, WC, and SMI (*p* < 0.05; Table 2).

#### 3.2.2. Biochemical and Blood Pressure Variables

Triglyceride levels were significantly different between the two groups (*p* < 0.05), whereas no significant differences were observed in fasting glucose or total cholesterol levels (*p* > 0.05; Table 3). Additionally, significant disparities were identified in both systolic blood pressure and diastolic blood pressure (*p* < 0.05), as shown in Table 3.

#### 3.2.3. Behavioral Habits

Tobacco and alcohol use were not statistically significant between groups (*p* > 0.05). The detailed outcomes of tobacco and alcohol consumption are presented in Table 4.

### 3.3. Odds Ratio for Sarcopenic Obesity

Table 5 demonstrates odds ratios with confidence intervals (CIs) for sarcopenic obesity in males, according to multiple logistic regression analysis. Statistical significance was observed at *p* < 0.05 for height (0.78, 95% CI: 0.683–0.854), weight (1.324, 95% CI: 1.214–1.523), WC (1.529, 95% CI: 1.354–1.726), triglycerides (1.117, 95% CI: 1.061–1.256), SBP (1.206, 95% CI: 1.124–1.295), and DBP (1.021, 95% CI: 1.012–1.045).

## 4. Discussion

This study assessed the prevalence of and risk factors associated with sarcopenic obesity among young-old males residing in the community. With the rapidly aging population in Korea and Asia in general, the incidence of sarcopenic obesity, particularly among males, is increasing. Despite the significant negative impact of sarcopenic obesity, healthcare professionals and primary care physicians face challenges in diagnosing the condition due to limited knowledge and diagnostic tools, resulting in overlooked diagnoses and complications. By utilizing variables such as age, height, weight, BMI, waist circumference, skeletal muscle index, smoking and drinking status, fasting glucose, triglycerides, total cholesterol, and systolic and diastolic blood pressure, this study employed a cost-effective, convenient, and accessible approach to identify individuals with potential sarcopenic obesity. Recognizing these risk factors is crucial for the early detection and prevention of sarcopenic obesity. Among males, the identified risk factors included waist circumference, skeletal muscle index, triglyceride levels, systolic blood pressure, and diastolic blood pressure.

Somatometric measures, specifically waist circumference, stand out as risk factors for age-related muscle loss, and various studies have indicated their association with an increased risk of sarcopenic obesity in men [18,19,20]. Brown et al. [18] conducted a study on a U.S. cohort and identified WC as a risk factor for sarcopenic obesity in men. Similarly, a cohort study conducted among individuals with sarcopenic obesity in Brazil revealed an association with a larger waist circumference [19]. Another study in Japan, which focused on community-dwelling individuals, suggested that those with sarcopenic obesity exhibited larger waist circumferences than those without sarcopenic obesity [20].

The theoretical foundation for the observed increase in waist circumference in adults with sarcopenic obesity is the interconnected relationship between increased fat mass and diminished muscle mass [21]. Individuals with sarcopenic obesity often encounter challenges related to muscle power and function due to muscle loss, leading to reduced engagement in physical activities, such as difficulties in sit-to-stand maneuvers and walking extended distances both indoors and outdoors [22]. This decline in physical activity strongly correlates with a reduction in total daily energy expenditure and an increase in body fat stores. Notably, fat accumulation tends to concentrate in the visceral and abdominal regions, ultimately contributing to the expansion of waist volume [22]. Consequently, the correlation between diminished muscle mass and fat mass accumulation in sarcopenic obesity is bidirectional and reinforcing [23]. Thus, consistent evidence highlighting waist circumference as a risk factor for sarcopenic obesity underscores the necessity for a better understanding of the intricate interplay between muscle and fat mass.

Elevated triglyceride levels were identified as a factor contributing to sarcopenic obesity, consistent with findings from prior research on sarcopenic obesity [24,25,26]. In their examination of community-dwelling elderly individuals, Lu et al. observed significant differences in triglyceride levels between sarcopenic obesity and normal groups [24]. A study of sarcopenic obesity in China reported substantially increased triglyceride levels in a sarcopenic obesity group compared with the normal elderly population [25]. Additionally, a longitudinal study conducted in Korea highlighted that males with sarcopenic obesity exhibited higher triglyceride levels than those in the normal group [26].

Insulin resistance is a plausible underlying mechanism explaining the observed association between sarcopenic obesity and elevated triglyceride levels. The disruption of lipid metabolism by insulin resistance is pivotal in this context. Normally, insulin facilitates the uptake of fatty acids and glucose by adipose tissue. However, in the presence of insulin resistance, this regulatory process is impaired, leading to an increased release of fatty acids from adipose tissue into the bloodstream [27]. Skeletal muscle, a crucial primary repository responsible for storing approximately 80% of ingested glucose after meals, plays a vital role in preventing hyperglycemia in the bloodstream [28]. However, individuals with sarcopenic obesity often exhibit notable reduction in insulin sensitivity. This diminished insulin sensitivity manifests as a decreased capacity for glucose uptake by the skeletal muscles, which is attributed to lower proportions of type I muscle fibers and diminished capillary density susceptible to insulin action [29].

Moreover, fat accumulation in individuals with sarcopenia contributes to synthesis. This process is facilitated by the liver via lipogenesis, which is a pivotal step in the overall metabolic panorama [30]. The liver, a central organ in an intricate lipid metabolism network, responds discerningly to excess circulating fatty acids in the systemic environment. This response is characterized by the initiation of lipogenesis. Within this intricate biochemical pathway, the liver engages in triglyceride synthesis from fatty acids and glycerol molecules, representing a pivotal juncture in the overall metabolic panorama. Lipogenesis fundamentally embodies the molecular processes orchestrated within hepatic cells. This intricate choreography of enzymatic reactions unfolds within hepatocytes, culminating in the conversion of fundamental building blocks, fatty acids, and glycerol molecules into more intricate and storage-ready triglycerides. These enzymatic transformations transcend biochemical processes and reflect the meticulous regulation of the liver molecular machinery. Each enzymatic step is intricately controlled to ensure seamless triglyceride synthesis [31,32].

Hypertension, encompassing both systolic and diastolic blood pressure, is a discernible risk factor for sarcopenic obesity. This finding is consistent with those of previous research studies [24,33,34]. Yin et al. investigated 14,928 Chinese adults and revealed elevated SBP levels in the male sarcopenic obesity group compared to the normal male group. Similarly, DBP was higher in men with sarcopenic obesity [34]. Another study conducted in a Taiwanese community found that the sarcopenic obesity group exhibited higher SBP than the normal population. Furthermore, DBP was higher in a sarcopenic obesity group than in a normal group [24]. A representative sample from a British male cohort study consisting of 7735 participants, reported significantly higher DBP in the sarcopenic obesity group than in the normal group [33].

Numerous underlying explanations may account for the observed increases in SBP and DBP. First, there is an intricate interplay between muscle loss and metabolic alterations, which leads to reduced energy expenditure and physical inactivity. These factors contribute to insulin resistance and arterial stiffness [35,36,37]. Second, augmentation of visceral fat mass triggers an inflammatory response, resulting in the thickening of blood vessel walls, impediment of blood flow, and constriction of vascular passages [38]. The susceptibility of males to hypertension is further accentuated by lower skeletal muscle mass and higher adipose tissue content. This unique physiological composition makes these individuals more prone to developing hypertension [39]. Diminished muscle mass and accumulation of adipose tissue, particularly in the visceral region, may contribute to the elevated prevalence of hypertension in men with obesity and sarcopenia.

Understanding the intricate relationship between hypertension and sarcopenic obesity requires a comprehensive understanding of the multifaceted mechanisms involved. The link between muscle loss, metabolic alterations, and insulin resistance underscores the interconnectedness of the physiological processes that contribute to elevated blood pressure in individuals with sarcopenic obesity. Moreover, the inflammatory response triggered by increased visceral fat mass elucidates the pathophysiological processes leading to arterial stiffness and hypertension. The unique physiological characteristics of males, such as lower skeletal muscle mass and higher adipose tissue content, contribute to their increased vulnerability to hypertension in the context of sarcopenic obesity. In summary, these findings highlight the importance of exploring the underlying mechanisms linking hypertension and sarcopenic obesity, paving the way for targeted interventions and preventive measures to mitigate the impact of this multifaceted condition on cardiovascular health.

The present study exhibits notable strength in its focused investigation of risk factors in males within a representative population of young individuals. This age group holds particular significance as sarcopenia progresses rapidly, and complications typically commence in this demographic [40,41,42,43]. These findings provide valuable insights for the early detection and treatment of sarcopenia.

However, it is essential to acknowledge the inherent limitations of this study, which warrant attention in future research. First, despite the inclusion of a substantial sample size of 1060 participants with representative statistical weights, the inherent nature of the cross-sectional design may restrict the ability to definitively establish causality between the identified risk factors and sarcopenic obesity. Elevated triglyceride and total cholesterol levels are potential predictors of sarcopenia. The cross-sectional design introduced the possibility that sarcopenic obesity itself could influence blood test results. Therefore, further research is required to comprehensively elucidate the intricate relationship between these predictors and the development of sarcopenic obesity. Future studies should consider longitudinal or randomized case–control designs to enhance the robustness of the findings. Another limitation pertains to the omission of an examination for sarcopenic obesity, a condition characterized by low muscle mass and high body fat. The exclusion of sarcopenic obesity is particularly relevant, as it may alter total cholesterol and triglyceride levels. To facilitate a more nuanced interpretation of the results, future research should consider the potential influence of sarcopenic obesity on the identified metabolic parameters. Next, the present study did not adhere to the Definition and Diagnostic Criteria for Sarcopenic Obesity as outlined by the ESPEN and EASO Consensus Statement in 2022 [1]. Although, we referenced the National Institutes of Health Sarcopenia Project definition in the United States [14] and adhered to the World Health Organization guideline for obesity [15]. In addition, a significant limitation is the lack of data on physical activity and dietary intake. Physical inactivity and dietary intake are key risk factors for SO and likely confound the association between sarcopenia and blood pressure/lipid profile observed in this study. The next study warrants a comprehensive discussion in future analyses.

Finally, in this study, we did not assess the body fat component by DEXA measurement. Recognizing the potential significance of including this measurement could offer valuable insights into sarcopenic obesity [44,45]. Future studies should prioritize incorporating the measurement of the body fat component for a more comprehensive understanding.

## 5. Conclusions

This nationwide investigation represents the initial clinical exploration of both the risk factors and prevalence of sarcopenic obesity in young-old males. The weighted prevalence was 9.09%, with a corresponding confidence interval ranging from 7.20% to 11.41%. The clinical risk factors for sarcopenia include systolic blood pressure, diastolic blood pressure, and triglyceride levels. Recognizing both the prevalence and identified risk factors will provide healthcare professionals with an enhanced ability to identify and detect potential cases of sarcopenia among men. Nevertheless, further research is imperative to deepen our understanding of the relationship between these risk factors and sarcopenia, and to bolster the overall robustness of these findings. Longitudinal or randomized case–control study designs hold promise for unraveling the intricacies of this association.

## Figures and Tables

**Table 1 healthcare-12-00700-t001:** Prevalence of specific sarcopenic obesity.

	Sarcopenic Obesity	Normal	Total
	(n = 95)	(n = 965)	(n = 1060)
Un-weighted (%)	8.95	91.05	100
Weighted ^1^ (%)	9.09 (7.20–11.41)	90.91 (88.59–92.80)	100

^1^ Weighted values present the 95% confidence interval.

**Table 2 healthcare-12-00700-t002:** Somatometric factors.

	Sarcopenic Obesity (n = 95)	Normal (n = 965)	*p*
Age (years)	69.526 ± 2.775	69.239 ± 2.793	0.337
Height (cm)	162.742 ± 4.901	166.941 ± 5.044	0.000 **
Weight (kg)	73.373 ± 7.104	64.148 ± 9.169	0.000 **
BMI (kg/m^2^)	27.662 ± 1.817	22.961 ± 2.706	0.000 **
WC (cm)	97.226 ± 5.577	84.352 ± 8.369	0.000 **
SMI (kg/m^2^)	0.722 ± 0.053	0.896 ± 0.073	0.000 **

BMI, body mass index; WC, waist circumference; SMI, skeletal muscle mass index. The independent *t*-test was performed. ** *p* < 0.001.

**Table 3 healthcare-12-00700-t003:** Biochemical and blood pressure variables.

	Sarcopenic Obesity (n = 95)	Normal (n = 965)	*p*
FG (mg/dL)	110.451 ± 32.386	104.831 ± 26.78	0.062
Triglyceride (mg/dL)	193.176 ± 184.242	136.534 ± 90.234	0.000 **
TC (mg/dL)	183.769 ± 36.842	179.928 ± 34.522	0.315
SBP (mmHg)	132.347 ± 16.792	128.717 ± 16.93	0.046 *
DBP (mmHg)	79.179 ± 9.599	76.953 ± 9.641	0.032 *

FG, fasting glucose; TC, total cholesterol; SBP, systolic blood pressure; DBP, diastolic blood pressure. The independent *t*-test was performed. * *p* < 0.05, ** *p* < 0.001.

**Table 4 healthcare-12-00700-t004:** Behavioral habits.

	Sarcopenic Obesity (n = 95)	Normal (n = 965)	*p*
Drinking status (%) (current-/ex-/non-drinker)	47.87/32.98/19.15	57.23/26.21/16.56	0.209
Smoking status (%) (current-/ex-/non-smoker)	70.21/22.34/7.45	72.03/17.22/10.75	0.333

A chi-square test was performed.

**Table 5 healthcare-12-00700-t005:** Multiple logistic regression for odds ratios of sarcopenia.

Variables	Odds Ratio (95% of CI)	*p*
Height	0.782 (0.683–0.854)	0.041 *
Weight	1.324 (1.214–1.523)	0.000 **
Waist circumference	1.529 (1.354–1.726)	0.000 **
Triglyceride	1.117 (1.061–1.256)	0.000 **
Systolic blood pressure	1.206 (1.124–1.295)	0.000 **
Diastolic blood pressure	1.021 (1.012–1.045)	0.000 **

The 95% confidence interval for the odds ratio (OR) was determined using multiple logistic regression. The odds ratio for sarcopenic obesity was calculated with every 1 cm in height, every 1 kg in weight, every 1 cm in waist circumference, every 1 mg/dL in triglyceride levels, every 1 mmHg in systolic blood pressure, and every 1 mmHg in diastolic blood pressure. * *p* < 0.05, ** *p* < 0.001.

## Data Availability

All data were anonymized and can be downloaded at https://knhanes.kdca.go.kr/knhanes, accessed on 1 February 2024.

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
