# Peer review of "A Comprehensive Risk Factor Exploration: Korean Nationwide Cross-Sectional Study of Sarcopenia Obesity in Young-Old Males—Investigating the Prevalence, Somatometric, Biochemical, and Behavioral Traits"

_healthcare, 2024, doi:10.3390/healthcare12060700_

Round 1
Reviewer 1 Report
Comments and Suggestions for Authors
Dear Authors,
Thank you for your manuscript.
You wrote an interesting paper that aimed to assess the prevalence of sarcopenic obesity and identified associated factors in young- old adults aged 65-74 years.
The article touches on the very relevant and interesting topic of sarcopenic satiety.
A strength of the study is the large study group.
The authors presented inconspicuous predictions of obesity prevalence in Korea.
The manuscript is well written and is well organized.
Some comments and suggestions:
1. In the Study Population section - there is no information that only men were included in the study.
2. It is unfortunate that the authors did not measure body fat in addition to muscle tissue. in the subjects. It is worth including these measurements in future studies.
3. In table 1 and 2 and 3 N-group size is once in lower case and once in upper case - it is worth standardising.
4. In line 162 there is a reference in the manuscript to Table3 and there should be a reference to Table 4.
Kind regards,
Comments on the Quality of English Language
Minor editing of English language required.
In line 178, for example, the 'of' is not needed.
Author Response
At first, authors express their deep gratitude to the reviewer’s valuable comments. know it is arduous job for reviewing papers. It consumes lots of time and effort. We were able to learn the way to write paper correctly because of your delicate comments. Again. We really appreciate it with my whole heart. Please see a attached word file.
*Please find attached our revised manuscript with changed from the original version highlighted with green (Please click an author's note file).
- In the Study Population section - there is no information that only men were included in the study.
Authors response: Thank you very much for your corrections. The authors feel sorry for the omission that men were included in the study. We amended it (lines 11-12, page 1; lines 68-69, page 2).
- It is unfortunate that the authors did not measure body fat in addition to muscle tissue. in the subjects. It is worth including these measurements in future studies.
Authors response: Authors entirely agree with the reviewer’s comment. We added the comment in the limitation part (lines 310-313, page 8). We will consider body fat component in the future study.
- In table 1 and 2 and 3 N-group size is once in lower case and once in upper case - it is worth standardising.
Authors response: Authors fully agree with the reviewer’s comment. We changed it (Table 1 and 2, 3 and 4, page 4-5)
- In line 162 there is a reference in the manuscript to Table3 and there should be a reference to Table 4.
Authors response: Authors totally agree with the reviewer’s comment. Thank you very much for your correction. The authors feel sorry for a wrong table number. We changed it (line 165, page 4).
Authors sincerely express their gratitude to the reviewer’s valuable comments, which have improved the readability and quality of this paper a lot.

Reviewer 2 Report
Comments and Suggestions for Authors
This study examined sarcopenia prevalence and risk factors for those between 65-74 in Korea. The main issues are methodological, including outdated definitions of sarcopenic obesity, issues with multiple comparisons, and the lack of considerations for confounders (most importantly physical activity data). These points will need to be justified/the paper revised prior to acceptance.
Methods: “253 individuals were classified as having both 80 sarcopenia and obesity, while 2,144 were classified as normal.” – this is different from the values in table 1. Please clarify.
The definition of sarcopenic obesity/sarcopenia has now shifted significantly – Please see donini 2022, Definition and Diagnostic Criteria for Sarcopenic
Obesity: ESPEN and EASO Consensus Statement. Please clarify why the diagnostic procedures were not followed.
Table 1 – what were the variables used to generate the weighted % of SO?
Table 2 – Sex needs to be in this table
Table 4 – the SBP finding does not make sense. For every unit increase of SBP, odds of sarcopenia decreased by 0.633. This does not make sense, and contradicts table 3, where SBP was higher in the sarcopenia group.
By including height weight and BMI in the same model you are likely to run into multicollinearity.
Please clarify the unit of measurement in table 4. Was it every 10mmhg? It would seem unlikely that every 1mmhg increase in DBP increased odds of sarcopenia by 8.6.
Consider correcting for multiple comparisons such as with a Bonferroni’s.
Why was table 5 analysis only done in males?
A major limitation is the lack of physical activity data. Physical inactivity is the leading modifiable risk factor for sarcopenia, and in fact likely to be a major confounder in the association between sarcopenia and blood pressure/lipid profile as reported in this study. See Foong et al, Accelerometer-determined physical activity, muscle mass, and leg strength in community-dwelling older adults, Journal of Cachexia, Sarcopenia and Muscle 2015, which demonstrates a dose-response relationship with physical activity and muscle mass/strength. Similarly, physical activity has a dose response relationship with obesity, so the exact same point as above applies. This needs to be discussed thoroughly.
The same as above goes for dietary intake.
Comments on the Quality of English Language-
Author Response
At first, authors express their deep gratitude to the reviewer’s valuable comments. know it is arduous job for reviewing papers. It consumes lots of time and effort. We were able to learn the way to write paper correctly because of your delicate comments especially in statistical analysis.. We really appreciate it with my whole heart. Please see a attached word file.
*Please find attached our revised manuscript with changed from the original version highlighted with yellow (Please click an author's note file).
This study examined sarcopenia prevalence and risk factors for those between 65-74 in Korea. The main issues are methodological, including outdated definitions of sarcopenic obesity, issues with multiple comparisons, and the lack of considerations for confounders (most importantly physical activity data). These points will need to be justified/the paper revised prior to acceptance.
- Methods: “253 individuals were classified as having both 80 sarcopenia and obesity, while 2,144 were classified as normal.” – this is different from the values in table 1. Please clarify.
Authors response: Thank you very much for your corrections. The authors really feel sorry for The mistake in the study. We corrected it (lines 80-81, page 2).
- The definition of sarcopenic obesity/sarcopenia has now shifted significantly – Please see donini 2022, Definition and Diagnostic Criteria for Sarcopenic Obesity: ESPEN and EASO Consensus Statement. Please clarify why the diagnostic procedures were not followed.
Authors response: The authors regret not adhering to the Definition and Diagnostic Criteria for Sarcopenic Obesity as outlined in the ESPEN and EASO Consensus Statement. Although we utilized the National Institutes of Health Sarcopenia Project definition in the United States (lines 95-96, page 3) and followed the World Health Organization guideline for obesity (line 100, page 3). We acknowledge that our study did not follow the established criteria for Sarcopenic Obesity outlined by ESPEN and EASO. We have addressed this limitation in the discussion section, highlighting the deviation from the recommended criteria for Sarcopenic Obesity. (lines 302-306, page 8)
- Table 1 – what were the variables used to generate the weighted % of SO?
Authors response: We apologize for making you confused. The weighted % of SO was analyzed by complex sampling analysis.
To be specific, in order to reflect the whole population in Korea, weighted values were applied to each sample through a three-step process: (1) calculation of the base weight, (2) adjustment for non-responses, and (3) post-stratification adjustment to match the entire previous census population. All analyses were conducted using complex sampling, with each individual adjusted for the weights provided by the KNHANES
All statistics of this survey have been calculated using sample weights assigned to sample participants. The sample weights were constructed for sample participants to represent the Korean population by accounting for the complex survey design, survey non-response and post-stratification. The weights based on the inverse of selection probabilities and inverse of response rates were modified by adjusting them to the sex- and age-specific Korean populations (post-stratification).
- Table 2 – Sex needs to be in this table
Authors response: We apologize for any confusion caused. Only male subjects are included in the study. The following reasons explain why we have chosen to include only male subjects:.
Males have a higher incidence of age-related loss of skeletal muscle mass than females (Bouchard, Dionne, & Brochu, 2009a; Dufour, Hannan, Murabito, Kiel, & McLean, 2013). Bouchard et al. (Bouchard, Dionne, & Brochu, 2009b) examined a group of 904 Canadians. They concluded that the incidence of CALSMO in men and women was 19% and 11%, respectively. Defour et al. (Dufour et al., 2013) carried out a study on a cohort of 767 individuals in the US as part of the Framingham study. They reported that the prevalence rate of SO was 8% and 4% among men and women, respectively. In other words, men are more prone to experiencing SO associated with aging than women. Identifying the risk factors and managing SO among older adults, with a particular emphasis on the significant proportion of affected males, is a challenge. These challenges are evident when comparing the existing research on SO in males to the well-researched studies focused on SO in females (Choi et al., 2022; Petroni et al., 2019; Rossi et al., 2019).
Therefore, we focused on males to investigate the risk factors associated with sarcopenic obesity.
- Table 3 – the SBP finding does not make sense. For every unit increase of SBP, odds of sarcopenia decreased by 0.633. This does not make sense, and contradicts table 3, where SBP was higher in the sarcopenia group.
Authors response: Authors sincerely apologize for the errors present in Table 5 of our manuscript. Regrettably, during the process of copying and pasting values, several mistakes occurred, resulting in inaccuracies within the table. We understand the importance of precise and reliable data presentation in academic literature and take full responsibility for these oversights.
To rectify these errors, we have thoroughly reviewed the data and made the necessary corrections. The revised Table 5 now accurately reflects the intended information, ensuring coherence and integrity in our presentation. Additionally, we have taken measures to enhance the content by providing further context where necessary, thereby enriching the reader's understanding (line 172, page 5, and table 5).
- By including height weight and BMI in the same model you are likely to run into multicollinearity.
Authors response: The authors wholeheartedly agree with the reviewer's insightful comment. We sincerely appreciate the correction. After careful consideration, we acknowledge that including variables such as height, weight, and BMI in the same model may lead to multicollinearity issues, thereby compromising the accuracy of our results. To address this concern, we have thoroughly reassessed our analysis process and decided to remove BMI from the odds ratio analysis. Consequently, all odds ratio values have been adjusted accordingly. We express our gratitude for your invaluable corrections once again (lines 169-175, page 5).
- Please clarify the unit of measurement in table 4. Was it every 10mmhg? It would seem unlikely that every 1mmhg increase in DBP increased odds of sarcopenia by 8.6.
Authors response: Thank you so much for your suggestion. Above mentioned, we changed oddratio thoroughly (lines 169-175, page 5).
- Consider correcting for multiple comparisons such as with a Bonferroni’s.
Authors' Response: The authors deeply appreciate the valuable feedback provided by the reviewer and wholeheartedly agree with their insightful comment. We also express regret for not implementing Bonferroni's correction to address potential errors. To rectify this oversight, we have recalculated the p-values using Bonferroni's correction method, as detailed in the methods section (lines 137-139, page 3). Additionally, the amended p-values are reflected in Table 5 (Table 5, page 5).
We are grateful for the reviewer's guidance in enhancing the rigor of our study.
- Why was table 5 analysis only done in males?
Authors response: We apologize for making you confused. We mentioned in the number 4 but we rewrite again for your convenience. Only male subjects are included in the study. The following reasons explain why we have chosen to include only male subjects:
Males have a higher incidence of age-related loss of skeletal muscle mass than females (Bouchard, Dionne, & Brochu, 2009a; Dufour, Hannan, Murabito, Kiel, & McLean, 2013). Bouchard et al. (Bouchard, Dionne, & Brochu, 2009b) examined a group of 904 Canadians. They concluded that the incidence of CALSMO in men and women was 19% and 11%, respectively. Defour et al. (Dufour et al., 2013) carried out a study on a cohort of 767 individuals in the US as part of the Framingham study. They reported that the prevalence rate of SO was 8% and 4% among men and women, respectively. In other words, men are more prone to experiencing SO associated with aging than women. Identifying the risk factors and managing SO among older adults, with a particular emphasis on the significant proportion of affected males, is a challenge. These challenges are evident when comparing the existing research on SO in males to the well-researched studies focused on SO in females (Choi et al., 2022; Petroni et al., 2019; Rossi et al., 2019).
Therefore, we focused on males to investigate the risk factors associated with sarcopenic obesity.
- A major limitation is the lack of physical activity data. Physical inactivity is the leading modifiable risk factor for sarcopenia, and in fact likely to be a major confounder in the association between sarcopenia and blood pressure/lipid profile as reported in this study. See Foong et al, Accelerometer-determined physical activity, muscle mass, and leg strength in community-dwelling older adults, Journal of Cachexia, Sarcopenia and Muscle 2015, which demonstrates a dose-response relationship with physical activity and muscle mass/strength. Similarly, physical activity has a dose response relationship with obesity, so the exact same point as above applies. This needs to be discussed thoroughly. The same as above goes for dietary intake.
Authors response: The authors fully concur with the reviewer’s insightful comment regarding the necessity to consider physical activity and dietary intake in the study. Acknowledging this crucial aspect, we have included a discussion of this limitation in the discussion section of our manuscript. (lines 306-309, page 8)
Thank you very much for the reviewer’s so many valuable comments. Thanks to the reviewer, the authors have improved their understanding of this research contents and the quality of the paper a lot. Once again, thank you.

Round 2
Reviewer 2 Report
Comments and Suggestions for Authors
Thank you for the comprehensive response - the authors have addressed many of the concerns.
Additional comments:
Methods -
"After excluding 35,736 participants outside female and aged 76 less than 65 and more than74 years old male, 1,837 participants were included in the study. 77 Of these, 777 were excluded because they did not receive dual-energy X-ray absorptiom- 78 etry (DEXA) examination or did not respond to the health survey, leaving a final sample 79 of 1,060 participants aged 65–74 years. " - this sentence does not read well and needs correction.
Please clarify the unit of measurement in table 5. Was it every 10mmhg for BP, every 10 mg/dL? this needs to be clarified in the table.
Lines 302 - 309 is a good addition to the limitations - however the citations seem to be missing.
Lines 310 - 313 is a good addition to limitations. Objective measure of body fat is useful and superior to simple measures such as BMI. Please find a study to cite here.
Comments on the Quality of English Language
-
Author Response
At first, authors express their deep gratitude to the reviewer’s valuable comments again. We really appreciate it with my whole heart.
*Please find attached our revised manuscript with changed from the original version highlighted with Truquois (Please click an author's note file).
- Methods - "After excluding 35,736 participants outside female and aged 76 less than 65 and more than74 years old male, 1,837 participants were included in the study. 77 Of these, 777 were excluded because they did not receive dual-energy X-ray absorptiom- 78 etry (DEXA) examination or did not respond to the health survey, leaving a final sample 79 of 1,060 participants aged 65–74 years. " - this sentence does not read well and needs correction.
Authors' Response: The authors completely agree with the reviewer's comment concerning the necessity of readability and coherence of sentences. Acknowledging this critical aspect, we have revised it accordingly (lines 76-80, page 2).
- Please clarify the unit of measurement in table 5. Was it every 10mmhg for BP, every 10 mg/dL? this needs to be clarified in the table.
Authors' Response: We sincerely appreciate the reviewer's comment. We have made the necessary revisions to clarify the unit of measurement in table 5(). These clarification aim to enhance the understanding of the data presented in the table 5 (lines 177-179, page 5).
- Lines 302 - 309 is a good addition to the limitations - however the citations seem to be missing.
Authors' Response: Thank you for your suggestion we added the citation (line 306, page 7).
Donini, L.M.; Busetto, L.; Bischoff, S.C.; Cederholm, T.; Ballesteros-Pomar, M.D.; Batsis, J.A.; Bauer, J.M.; Boirie, Y.; Cruz-Jentoft, A.J.; Dicker, D. Definition and diagnostic criteria for sarcopenic obesity: ESPEN and EASO consensus statement. Obesity Facts 2022, 15, 321-335.
- Lines 310 - 313 is a good addition to limitations. Objective measure of body fat is useful and superior to simple measures such as BMI. Please find a study to cite here.
Authors' Response: Thank you for your suggestion we added the citation (line 306, page 7).
Fukuda, T.; Bouchi, R.; Takeuchi, T.; Tsujimoto, K.; Minami, I.; Yoshimoto, T.; Ogawa, Y. Sarcopenic obesity assessed using dual energy X-ray absorptiometry (DXA) can predict cardiovascular disease in patients with type 2 diabetes: a retrospective observational study. Cardiovasc. Diabetol. 2018, 17, 55, doi:10.1186/s12933-018-0700-5.
Liu, C.; Cheng, K.Y.-K.; Tong, X.; Cheung, W.-H.; Chow, S.K.-H.; Law, S.W.; Wong, R.M.Y. The role of obesity in sarcopenia and the optimal body composition to prevent against sarcopenia and obesity. Front. Endocrinol. (Lausanne) 2023, 14, 1077255.
Thank you very much for the reviewer’s so many valuable comments. Thanks to the reviewer, the authors have improved their understanding of this research contents and the quality of the paper a lot. Once again, thank you.
